# Vitamin D Intake May Reduce SARS-CoV-2 Infection Morbidity in Health Care Workers

**DOI:** 10.3390/nu14030505

**Published:** 2022-01-24

**Authors:** Tatiana L. Karonova, Alena T. Chernikova, Ksenia A. Golovatyuk, Ekaterina S. Bykova, William B. Grant, Olga V. Kalinina, Elena N. Grineva, Evgeny V. Shlyakhto

**Affiliations:** 1Clinical Endocrinology Laboratory, Department of Endocrinology, Almazov National Medical Research Centre, 194021 Saint Petersburg, Russia; arabicaa@gmail.com (A.T.C.); ksgolovatiuk@gmail.com (K.A.G.); bykova160718@gmail.com (E.S.B.); olgakalinina@mail.ru (O.V.K.); grineva_e@mail.ru (E.N.G.); shlyakhto_ev@almazovcentre.ru (E.V.S.); 2Sunlight, Nutrition, and Health Research Center, San Francisco, CA 94164-1603, USA; williamgrant08@comcast.net

**Keywords:** COVID-19, SARS-CoV-2, vitamin D, 25(OH)D, health care workers

## Abstract

In the last 2 years, observational studies have shown that a low 25-hydroxyvitamin D (25(OH)D) level affected the severity of infection with the novel coronavirus (COVID-19). This study aimed to analyze the potential effect of vitamin D supplementation in reducing SARS-CoV-2 infection morbidity and severity in health care workers. Of 128 health care workers, 91 (consisting of 38 medical doctors (42%), 38 nurses (42%), and 15 medical attendants (16%)) were randomized into two groups receiving vitamin D supplementation. Participants of group I (*n* = 45) received water-soluble cholecalciferol at a dose of 50,000 IU/week for 2 consecutive weeks, followed by 5000 IU/day for the rest of the study. Participants of group II (*n* = 46) received water-soluble cholecalciferol at a dose of 2000 IU/day. For both groups, treatment lasted 3 months. Baseline serum 25(OH)D level in health care workers varied from 3.0 to 65.1 ng/mL (median, 17.7 (interquartile range, 12.2; 24.7) ng/mL). Vitamin D deficiency, insufficiency, and normal vitamin D status were diagnosed in 60%, 30%, and 10%, respectively. Only 78 subjects completed the study. Vitamin D supplementation was associated with an increase in serum 25(OH)D level, but only intake of 5000 IU/day was accompanied by normalization of serum 25(OH)D level, which occurred in 53% of cases. Neither vitamin D intake nor vitamin D deficiency/insufficiency were associated with a decrease in SARS-CoV-2 morbidity (odds ratio = 2.27; 95% confidence interval, 0.72 to 7.12). However, subjects receiving high-dose vitamin D had only asymptomatic SARS-CoV-2 in 10 (26%) cases; at the same time, participants who received 2000 IU/day showed twice as many SARS-CoV-2 cases, with mild clinical features in half of them.

## 1. Introduction

Recent studies showed that vitamin D deficiency and insufficiency are common in the world’s general population, including Russia [1,2]. The COVID-19 pandemic that swept the world in 2019 changed our lifestyles, as well as scientific and medical approaches. SARS-CoV-2, a respiratory viral airborne infection with no effective treatment other than applying prevention strategies, had serious damaging health effects [3]. In the last 2 years, researchers have shown that COVID-19 severity may be related to vitamin D status [4]. Considering vitamin D’s immunomodulatory properties [5,6,7,8], scientists found associations between low 25(OH)D level and COVID-19 severity in observational studies [9,10,11,12]. The largest study in the United States reported that the SARS-CoV-2 positivity rate was 6%. It was lowest in subjects with 25(OH)D concentration >55 ng/mL, compared with patients with 25(OH)D of 20 ng/mL, whose rate reached about 11% [13]. However, for hospitalized patients, being aged 50 years or older was a more significant factor than vitamin D status [14]. At the same time, we published data with analysis for more than 300,000 subjects who had a known 25(OH)D level from fall 2019 to fall 2020, and detected that vitamin D deficiency did not increase the rate of positive PCR test to SARS-CoV-2 in the Russian population [15]. One reason for that difference between the United States and Russia could be the high prevalence of vitamin D deficiency in Russia, as well as using different methods for the PCR testing of COVID-19.

Health care workers are now considered a risk group because they are continually exposed to COVID-19. In a study of 120,075 participants, Mutambudzi and colleagues showed that medical support staff had a sevenfold-higher risk of severe COVID-19 infection than other worker groups (relative risk (RR) = 7.43; 95% confidence interval (CI), 5.52 to 10.00) [16]. The same data were found in the United Kingdom, where medical staff had a high risk of COVID-19, especially those with vitamin D deficiency [17].

Vitamin D activates immune cells to produce cathelicidin and defensins, as well as increasing expression of angiotensin-converting enzyme 2, which promotes the binding of the virus in the lung blood vessels [18], leading to reduced survival and replication. As a result, vitamin D supplementation could prevent and treat SARS-CoV-2. Results of the pilot study in Spain, in which hospitalized COVID-19-positive patients took calcifediol, showed that only 1 of 50 treated patients required the intensive-care unit in comparison with 13 of 26 nontreated patients (odds ratio (OR) = 0.02; 95% CI, 0.002 to 0.17) [19]. The bolus vitamin D supplementation also was associated with decreased mortality in hospitalized COVID-19 patients in Turkey [20], and an improved 3-month survival in geriatric patients [21]. This information was confirmed in a meta-analysis that included data from observational and randomized controlled trials that reported reduced severity risk with higher 25(OH)D, and some benefit from vitamin D in treating COVID-19 [22].

Despite many COVID-19 studies having been performed in the last 18 months, we could not find interventional studies among medical staff with the use of vitamin D supplementation to assess its preventive effect on SARS-CoV-2 morbidity. So, this study aimed to analyze the potential effect of vitamin D supplementation in reducing SARS-CoV-2 infection morbidity and severity in health care workers.

## 2. Materials and Methods

### 2.1. Patients

The study population included 128 employees of Almazov National Medical Research Centre (111 women and 17 men) who signed an informed consent for participation. They began work in the infectious hospital amid the COVID-19 pandemic, had a negative PCR test, and were off vitamin D or received only preventive doses. This single-center, open-label, randomized, interventional study was performed from October 30 2020 to February 28 2021, with the following inclusion criteria: (i) age 18–65 years, (ii) negative PCR test for SARS-CoV-2, (iii) absence of clinical signs of acute respiratory viral infection (ARVI), (iv) contact with patients with laboratory and/or clinically confirmed SARS-CoV-2 infection. We did not include subjects with a history of intolerance or allergic response to water-soluble cholecalciferol in anamnesis, or those not compliant with the recommendation of the Ministry of Health with regard to personal protective equipment [23]. Exclusion criteria included primary hyperparathyroidism or hypercalcemia of other etiology (including a mutation of 24 hydroxylase); clinically significant gastrointestinal diseases, kidney pathology (estimated glomerular filtration rate less than 45 mL/min/1.73 m^2^), and liver diseases that can influence vitamin D absorption and metabolism; a history of granulomatous diseases; a history of oncology diseases (<5 years); intake of glucocorticosteroids or anticonvulsants; and alcohol and drug addiction. Pregnant, breastfeeding women or women planning pregnancies also did not participate. If potential participants had other circumstances that the investigator considered inappropriate, they were excluded. Of note, the general vaccination, including for high-risk groups, was launched only in the end of February to early March 2021, and the first participant was included in the study in November 2020, when the center started to work with COVID-19 patients. Hence, we did not include vaccinated health care workers at the beginning or throughout the study.

After signing the informed consent and initial physical examination, 23 subjects (18%) were excluded from the survey after the prompt onset of a respiratory tract infection before taking the first dose of cholecalciferol. Eleven employees (9%) withdrew consent soon after signing; and three subjects (2%) were excluded because of a lack of initial laboratory data. Thus, the final survey included data for 91 health care workers. After randomization (random numbers method) at a ratio of 1:1, all participants were divided into two groups. The participants of group I (*n* = 45) received water-soluble cholecalciferol (Aquadetrim, “Akrichin,” Staraya Kupavna, Moscow region, Russia) at a dose of 50,000 IU/week for 2 consecutive weeks, followed by 5000 IU/day for the rest of the study. The participants of group II (*n* = 46) received water-soluble cholecalciferol (Aquadetrim) at a dose of 2000 IU/day. Our center was involved in the treatment of COVID-19 patients only for a duration of three months; hence, the health care workers had direct contact with such patients only during this time. Thus, the treatment lasted 3 months for both groups (Figure 1).

### 2.2. Physical Data

Anthropometric examination included height (centimeters) and weight (kilograms), from which body mass index (BMI) was calculated (kilograms per square meter of body surface area). The participant questionnaire included demographic data, education, medical history and concomitant medication, smoking status, allergies, and vitamin D supplement intake.

### 2.3. Laboratory Tests

Serum 25(OH)D level was detected by the chemiluminescent immunoassay (Architect i8000; Abbott, Chicago, IL, USA) using laboratory sets and control sera from the manufacturer. Vitamin D deficiency was defined as a serum 25(OH)D level < 20 ng/mL [24].

Testing of immunoglobulin G (IgG) to SARS-CoV-2 was performed with a semiquantitative method by using the enzyme-linked immunosorbent assay on the Bio-Rad 680 microplate reader equipment (Hercules, CA, USA) with the corresponding set SARS-CoV-2-IgG-ELISA-Best (Vector Best; Novosibirsk, Russia). A result was considered negative for positivity index (PI) < 0.8; positive for PI ≥ 1.1; and borderline for 0.8 ≤ PI < 1.1.

In addition, we evaluated biochemical parameters, such as a fasting plasma glucose level and blood lipid profile (Roche Diagnostics GmbH, Mannheim, Germany). Also, 2 weeks after starting the study, participants underwent blood tests to assess serum 25(OH)D (Abbott Architect i8000) and total calcium levels (reference interval, 2.15–2.65 mmol/L; Roche Diagnostics GmbH, Mannheim, Germany).

All blood samples were taken in the morning from the cubital vein, centrifuged, aliquoted, and stored in a freezer at −70 °C before testing.

### 2.4. Statistical Analysis

For sample calculation, we used Power and Sample Size software [25]. At a 5% significance level and 80% power, the sample size was 72 people (36 per group).

Statistical processing was carried out using SPSS for Windows (ver. 26; IBM, Armonk, NY, USA), with the help of standard methods of variation statistics. Between-group comparison was carried out using the Mann–Whitney criteria for incorrect distribution; results are presented as median and interquartile range, as well as mean and standard deviation for the Student criterion in correct distributed parameters. Associations between quantitative parameters were assessed using Spearman’s correlation coefficient. To describe relative risk, we calculated the odds ratio, with a 95% confidence interval calculated using Fisher’s exact method. The criterion for the statistical reliability of the obtained results was *p* < 0.05.

## 3. Results

Of the 128 health care workers who signed the informed consent, 57 were medical doctors (44%), 52 were nurses (41%), and 19 were medical attendants (15%). Baseline serum 25(OH)D level varied from 3.0 to 69.0 ng/mL (mean, 18.5 (interquartile range, 11.9; 26.7) ng/mL). A total of 114 participants presented with baseline 25(OH)D results, with 63 subjects (55%) deficient, 34 subjects (30%) insufficient, and only 17 health care workers (15%) with normal vitamin D status. Medical attendants were diagnosed with vitamin D deficiency more often than medical doctors and nurses: 88%, 46%, and 53%, respectively (*p* = 0.001). The participants with graduate medical education had a higher serum 25(OH)D level (22.1 (16.1; 29.5) ng/mL) than subjects with a secondary medical education (19.3 (10.7; 24.9) ng/mL) or without specialized education (11.1 (9.7; 17.6) ng/mL) (*p* = 0.001; Table 1).

After exclusion of subjects infected with COVID-19 or those who withdrew consent before the first dose of vitamin D supplementation, 91 employees (38 medical doctors, 38 nurses, and 15 medical attendants) were randomized. Participants in the groups were comparable and had no significant differences in baseline serum 25(OH)D level, which was 16.9 (11.4; 23.9) for group I, and 18.4 (12.2; 25.1) ng/mL for group II (*p* = 0.54; Table 2).

Two weeks after the initiation of vitamin D supplementation, serum 25(OH)D level and total calcium level were measured to control efficacy and safety. So, after 100,000 IU of water-soluble cholecalciferol after 2 weeks, median 25(OH)D level was 32.9 (26.3; 39.6) ng/mL, and was significantly higher than in the initial data (*p* = 0.001). Participants who received 2000 IU had no significant changes in 25(OH)D level (19.3 (14.1; 27.2) ng/mL); *p* = 0.08. Total serum calcium level was within reference values in both groups.

Analyses of IgG to SARS-CoV-2 showed that 13 randomized participants (14%) had initially positive IgG titers, an indicator of past infection—probably asymptomatic. Therefore, data of those participants were excluded from the final analysis.

The analyzable final results included data of 78 employees (34 medical doctors (44%), 33 nurses (42%), and 11 medical attendants (14%)) who had not been exposed to the virus in the past as a result of SARS-CoV-2 (Table 3). The results of their examination showed the absence of significant differences, including the values of the baseline serum 25(OH)D level in group I (18.4 (14.3; 24.5) ng/mL) and group II (18.5 (12.5; 25.0) ng/mL) (*p* = 0.94).

All participants had an increase in serum 25(OH)D level at the end of the study. Therefore, serum 25(OH)D after 3 months of vitamin D supplementation reached 29.9 (25.2; 42.0) ng/mL in group I and 26.0 (21.3; 30.3) ng/mL in group II (*p* = 0.01), with 53% of participants from group I and 25% from group II reaching normal vitamin D status (Figure 2).

Analysis of positive IgG to SARS-CoV-2 cases among employees showed that 10 (26%) health care workers in group I had a positive PCR test and positive IgG titer, but no clinical features of ARVI. At the same time, 18 (45%) employees in group II had positive results, including 9 (23%) subjects with mild ARVI clinical features, and 9 (23%) subjects with asymptomatic disease. No participants underwent a computed tomography scan owing to asymptomatic or a mild course of COVID-19. Baseline and following vitamin D supplementation serum 25(OH)D level was the same among participants with positive IgG to SARS-CoV-2, and virus-free participants (baseline, 19.3 (12.1; 23.6) and 16.9 (11.8; 24.9) ng/mL, *p* = 0.51; and, at end of the study, 26.4 (20.3; 29.3) and 27.2 (22.2; 36.4) ng/mL, *p* = 0.69, respectively). Assessing the risk of SARS-CoV-2 morbidity depending upon vitamin D status, we found no associations between vitamin D deficiency/insufficiency and increased incidence of viral infection (OR = 2.27; 95% CI, 0.72 to 7.12).

## 4. Discussion

To our knowledge, this is the first randomized interventional trial among health care workers to show that high-dose vitamin D supplementation is safe and effective in achieving normal vitamin D levels, but was not connected to reduced SARS-CoV-2 morbidity. However, intake of 50,000 IU/week twice, followed by 5000 IU/day, seemed to be associated with asymptomatic COVID-19 cases, whereas health care workers receiving 2000 IU/day had a two-fold higher infection that was symptomatic with mild clinical features in half of cases.

Vitamin D is postulated to play an important immunomodulatory role, and deficiency is associated with increased incidence of ARVI, including COVID-19 [26,27,28]. Our previous results also showed that severe vitamin D deficiency is associated with severity and death in COVID-19 patients [29], and were comparable to recent findings [4,30]. Dissanayake and colleagues, whose meta-analysis included 72 COVID-19 observational and 4 interventional randomized studies, have shown not only correlations between 25(OH)D level and severity or mortality, but also some clinical benefits and improvement in inflammatory markers of vitamin D supplementation in treating COVID-19 [22].

Recent observational studies showed a more frequent vitamin D deficiency among shift workers and newcomers, including health care workers, than day workers [17,31,32], whereas data regarding mortality rate showed a high COVID-19-related mortality among health care workers, as published by the World Health Organization [33]. Thus, a great necessity exists to find new effective measures to prevent SARS-CoV-2 and/or decrease COVID-19 morbidity and severity in medical workers. Taking into account the need to improve preventive actions for medical staff in daily contact with SARS-CoV-2 patients, we developed a hypothesis for this research to assess vitamin D supplementation’s effectiveness in preventing COVID-19 among this population.

To reduce the risk of infection, it is recommended that people at risk should rapidly increase 25(OH)D concentrations above 40–60 ng/mL [27,34]. In order to achieve this, patients need to take higher loading vitamin D doses: 100,000–200,000 IU over 8 weeks [34,35]. To maintain that level after the first month, the dose can be decreased to 5000 IU/day [36]. Considering published recommendations for decreasing the morbidity of COVID-19 [27,34,35], we have chosen a high vitamin D supplementation dose of 50,000 IU/week twice for a rapid increase of 25(OH)D level, followed by 5000 IU/day, and compared with the common daily dose used in clinical practice. We showed a good tolerability of the saturating dose of water-soluble cholecalciferol, and a rapid increase in the serum 25(OH)D level to normal values without an increase in total calcium levels in the blood. Those results are comparable to those of previous works [36,37].

Inspection of the population-based study results shows that subjects with cholecalciferol supplementation had a lower risk of SARS-CoV-2 infection (hazard ratio (HR) = 0.95 (95% CI, 0.91 to 0.98); *p* = 0.004) than deficient unsupplemented subjects. The protective effect was more significant between treated subjects with 25(OH)D > 30 ng/mL and untreated deficient subjects (HR = 0.57 (95% CI, 0.50 to 0.66); *p* < 0.001) [38]. In our study, no significant differences were evident in morbidity between the comparable groups, and no difference in serum 25(OH)D level emerged between subjects with positive or negative IgG titers despite vitamin D supplementation. That result might be related to our inability to achieve the recommended 25(OH)D level of 40–60 ng/mL. However, subjects receiving a higher dose of cholecalciferol had an asymptomatic course of viral infection. Those differences can be explained by engagement mechanisms as in the cell-bound and adaptive immunity, as well as a protective function on the level of upper-airway mucosa [39].

Possible study limitations include the small sample, absence of lab baseline data of serum 25(OH)D level and IgG before randomization, and short study duration of 3 months. In addition, the study was carried out in a comparative rather than placebo-controlled design. Therefore, conducting more detailed research is necessary to better understand vitamin D’s role in preventing SARS-CoV-2 infection.

## Figures and Tables

**Figure 1 nutrients-14-00505-f001:**
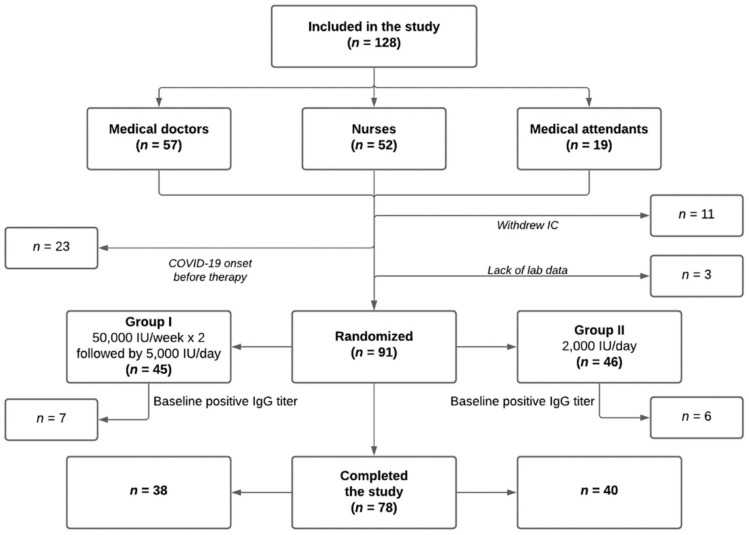
Study design. IC, informed consent; IgG, immunoglobulin G.

**Figure 2 nutrients-14-00505-f002:**
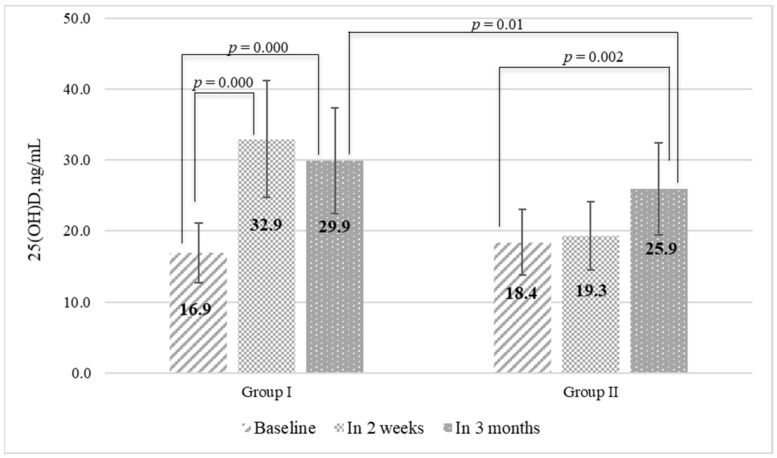
Serum 25(OH)D level before and after different doses of vitamin D supplementation.

**Table 1 nutrients-14-00505-t001:** Vitamin D status among health care workers (*n* = 114).

Parameter	Medical Doctors *n* = 52	Nurses *n* = 45	Medical Attendants *n* = 17	*p*
25(OH)D, ng/mL, Me + IQR (25; 75)	22.1 (16.1; 29.5)	19.3 (10.7; 24.9)	11.1 (9.7; 17.6)	0.001
Vitamin D status, *n* (%)				
Normal	12 (23)	4 (9)	1 (6)	0.001
Insufficiency	16 (31)	17 (38)	1 (6)	
Deficiency	24 (46)	24 (53)	15 (88)	

25(OH)D, 25-hydroxyvitamin D; Me, median; IQR, interquartile range.

**Table 2 nutrients-14-00505-t002:** Baseline characteristics of randomized health care workers.

Parameters	Group I *n* = 45	Group II *n* = 46	*p*
Age, years (mean ± SD)	35 ± 2	35 ± 2	0.81
Sex, M/F, *n* (%)	8 (18)/37 (82)	6 (13)/40 (87)	0.53
Education, *n* (%)			
Graduate medical	15 (33)	23 (50)	0.36
Secondary medical	24 (53)	14 (30)	
Without specialized education	6 (14)	9 (20)	
BMI, kg/m^2^, *n* (%)	24.8 ± 0.8	24.6 ± 0.7	0.98
Normal	25 (55)	29 (63)	0.49
Overweight	12 (27)	10 (22)	
Obese	8 (18)	7 (15)	
FPG, mmol/L	5.3 ± 0.2	5.3 ± 0.2	0.35
TC, mmol/L	5.3 ± 0.2	5.3 ± 0.2	0.95
LDL, mmol/L	2.9 ± 0.2	3.0 ± 0.1	0.46
HDL, mmol/L	1.6 ± 0.1	1.6 ± 0.1	0.44
TG, mmol/L	1.6 ± 0.2	1.6 ± 0.2	0.49
25(OH)D, ng/mL, Me + IQR (25; 75)	16.9 (11.4; 23.9)	18.4 (12.2; 25.1)	0.54
Vitamin D status, *n* (%)			
Normal	4 (9)	5 (11)	0.45
Insufficiency	12 (27)	15 (33)	
Deficiency	29 (64)	26 (56)	

SD, standard deviation; M, male; F, female; BMI, body mass index; FPG, fasting plasma glucose; TC, total cholesterol; LDL, low-density lipoprotein; HDL, high-density lipoprotein; TG, triglycerides; 25(OH)D, 25-hydroxyvitamin D; Me, median; IQR, interquartile range.

**Table 3 nutrients-14-00505-t003:** Characteristics of health care workers with initially negative IgG titer to SARS-CoV-2.

Parameters	Group I *n* = 38	Group II *n* = 40	*p*
Age, years (mean ± SD)	34 ± 2	36 ± 2	0.93
Sex, M/F, *n* (%)	6 (16)/32 (84)	6 (15)/34 (85)	0.92
Education, *n* (%)			0.99
Graduate medical	15 (39)	19 (48)	
Secondary medical	20 (53)	13 (32)	
Without specialized education	3 (8)	8 (20)	
BMI, kg/m^2^, *n* (%)	24.3 ± 0.9	24.7 ± 0.7	0.57
Normal	22 (58)	25 (63)	0.85
Overweight	12 (32)	9 (22)	
Obesity	4 (10)	6 (15)	
FPG, mmol/L	4.9 ± 0.1	5.4 ± 0.2	0.12
TC, mmol/L	5.2 ± 0.2	5.3 ± 0.2	0.91
LDL, mmol/L	2.8 ± 0.2	3.0 ± 0.1	0.44
HDL, mmol/L	1.6 ± 0.1	1.6 ± 0.1	0.29
TG, mmol/L	1.5 ± 0.2	1.6 ± 0.2	0.81
25(OH)D, ng/mL, Me + IQR (25; 75)	18.4 (14.3; 24.5)	18.5 (12.5; 25.0)	0.94
Vitamin D status, *n* (%)			
Normal	3 (8)	5 (12)	
Insufficiency	12 (32)	12 (30)	
Deficiency	23 (60)	23 (58)	

SD, standard deviation; M, male; F, female; BMI, body mass index; FPG, fasting plasma glucose; TC, total cholesterol; HDL, high-density lipoprotein; LDL, low-density lipoprotein, TG, triglycerides; Me, median; IQR, interquartile range; 25(OH)D, 25-hydroxyvitamin D.

## Data Availability

The data generated and analyzed during this study are included in this article and its supplementary information files. More information is available from the corresponding author on reasonable request.

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
