# Peer review of "Vitamin D Intake May Reduce SARS-CoV-2 Infection Morbidity in Health Care Workers"

_nutrients, 2022, doi:10.3390/nu14030505_

Round 1

Reviewer 1 Report

In the publication Vitamin D Intake May Prevent SARS-CoV-2 Infection in 2 Health Care Workers the authors deal with the influence of the vitamin D on health  and the incidence of COVID-19 among health care personnel.

The publication has clear and consistent layout. Paragraphs are correct and the literature is carefully prepared, it is the most up-to-date

1.    A problem of incidences of  the COVID-19 for this group of people is very important issue as it affects the efficiency of the healthcare system. One can have serious concerns about the lack of a control group and that analysis were carried out on not to big group of people. In addition, the 3-month study time is also short to say something more about the effects on the body, even if we consider it in the sole purpose of SARS-CoV2. The authors explain all this in the last sentences of the article, but this is rather significant objection and makes the work to be treated rather as pilot studies.

  1. Moreover, it is not understood why information on vaccination is not included. The time of observation already applies to months, when medical personnel were vaccinated all over the world. It is important information to know whether the subjects were vaccinated or not? If medical personnel has not yet been vaccinated at all (while the analysis was being done), the work is so unique that it is impossible to repeat the results now when everyone is vaccinated
  2. Explaining who and why quit the research slightly disturbs the transparency of the work, but it may be a subjective feeling. Will it not enough to describe exactly which people, with what contraindications, were not taken into account. There is also a disproportion in the group between the number of men and women participating in the
  3. There is a cited literature in publication but it is worth to add an explanation why exactly the dose of 5,000 IU and 2000 IU and 50,000 IU  were selected.

Author Response

Dear Reviewer!

We would like to thank you for the constructive and helpful comments that you have kindly made with regard to the manuscript entitled Vitamin D Intake May Prevent SARS-CoV-2 Infection in Health Care Workersintended for publication in the Nutrients as an original research article.

Please find the revised manuscript and our responses to the comments. The changes made to the text have been highlighted.

Comments and Suggestions for Author

In the publication Vitamin D Intake May Prevent SARS-CoV-2 Infection in 2 Health Care Workers the authors deal with the influence of the vitamin D on health and the incidence of COVID-19 among health care personnel.

The publication has clear and consistent layout. Paragraphs are correct and the literature is carefully prepared, it is the most up-to-da

  1. A problem of incidences of the COVID-19 for this group of people is very important issue as it affects the efficiency of the healthcare system. One can have serious concerns about the lack of a control group and that analysis were carried out on not to big group of people. In addition, the 3-month study time is also short to say something more about the effects on the body, even if we consider it in the sole purpose of SARS-CoV2. The authors explain all this in the last sentences of the article, but this is rather significant objection and makes the work to be treated rather as pilot studies.

Answer: We agree that randomized controlled trials and large population studies should be conducted to evaluate such effects.

Our study included medical stuff that was working with patients who had confirmed SARS-CoV-2 in the Center during the 3-month period. Our center was involved in the treatment of COVID-19 patients only for three months, and hence the health care workers had direct contact with such patients only for this duration of time. We do not have a control group since our Ethics Committee did not approve the control group among health care workers in contact with patients who had confirmed SARS-CoV-2 infection, those being as a high-risk of SARS-Co-V-2 infection group. Thus, group II received cholecalciferol at the common daily dose used in clinical practice – 2,000 IU/day.

  1. Moreover, it is not understood why information on vaccination is not included. The time of observation already applies to months, when medical personnel were vaccinated all over the world. It is important information to know whether the subjects were vaccinated or not? If medical personnel has not yet been vaccinated at all (while the analysis was being done), the work is so unique that it is impossible to repeat the results now when everyone is vaccinated.

Answer:  The vaccination in Russia started in Dec 20-Jan 21, however general vaccination including the high-risk groups was launched only in the end of February early March. The study was approved  in October 30, 2020 and the first participant was included in the study in Nov, when the Centre started to work with COVID-19 patients. We did not include health care workers who underwent the vaccination from SARS-Co-V-2 infection at the beginning and throughout the study. We made corrections and added information about vaccination in the text.

  1. Explaining who and why quit the research slightly disturbs the transparency of the work, but it may be a subjective feeling. Will it not enough to describe exactly which people, with what contraindications, were not taken into account. There is also a disproportion in the group between the number of men and women participating in the study

Answer:  Thank you for your comment. But as you indicated earlier “that analysis was carried out on a small sample size” so we tried to explain exclusion of subjects in details. The disproportion in the groups is common for medical stuff in Russia, so the distribution we see in the study reflects the reality.

  1. There is a cited literature in publication but it is worth to add an explanation why exactly the dose of 5,000 IU and 2000 IU and 50,000 IU were selected.

Answer: As explained earlier the local Ethics committee did not allow placebo control trial so the second group received common daily dose used in clinical practice in pandemic period – 2,000 IU/day. The dose in 5,000 IU/day as mentioned in reviewed paper was chosen based on published recommendations and experience for rapid increase of 25(OH)D for the pleiotropic effect.

Reviewer 2 Report

Review of “Vitamin D Intake May Prevent SARS-CoV-2 Infection in Health Care Workers” by Karonova et al. 

This is an interesting study of whether vitamin D supplementation affects SARS-CoV-2 infection morbidity and severity in a group of 128 health care workers. Overall, it is well written, and the data seems supportive to the conclusion. Below I provide few comments for improve.

1. The title sounds a bit ambitious. The authors did a better job in the abstract by phrasing it as “reducing SARS-CoV-2 infection morbidity and severity” but not “prevent”. The title should be rephased accordingly.

2. Information of the vaccination status of these health care workers is missing. As health care workers is a high-risk group, one would assume that they (or a subgroup of them) have been vaccinated. Vaccination will affect the infection risk, therefore this information needs to be present in the study.

3. Why group II have a vitamin D supplementation of 2000IU/day? If the authors aim to explore benefit of Vitamin D, why not compare to a group that with no vitamin D supplementation? Current setting is more like a comparison between high and low vitamin D supplementation.

4. The Serum 25(OH)D level of these two groups seems mostly differ in the first two weeks (Figure 2).  Is there a significant difference of infection in this two-weeks period? 

Author Response

Dear Reviewer!

We would like to thank you for careful consideration of the manuscript entitled “Vitamin D Intake May Prevent SARS-CoV-2 Infection in Health Care Workers” intended for publication in the Nutrients as an original research article.

Please find the revised manuscript and our responses to the comments. The changes made to the text have been highlighted.

Comments and Suggestions for Authors

This is an interesting study of whether vitamin D supplementation affects SARS-CoV-2 infection morbidity and severity in a group of 128 health care workers. Overall, it is well written, and the data seems supportive to the conclusion. Below I provide few comments for improve.

  1. The title sounds a bit ambitious. The authors did a better job in the abstract by phrasing it as “reducing SARS-CoV-2 infection morbidity and severity” but not “prevent”. The title should be rephased accordingly.

Answer: We agree with this idea and we updated the title of our paper as per your advice Vitamin D Intake May Reducing SARS-CoV-2 Infection Morbidity in Health Care Workers

  1. Information of the vaccination status of these health care workers is missing. As health care workers is a high-risk group, one would assume that they (or a subgroup of them) have been vaccinated. Vaccination will affect the infection risk, therefore this information needs to be present in the study.

Answer: The vaccination in Russia started in Dec 20-Jan 21, however general vaccination including the high-risk groups was launched only in the end of February early March. The study was approved in October 30, 2020 and the first participant was included in the study in Nov, when the Centre started to work with COVID-19 patients. We did not include health care workers who underwent the vaccination from SARS-Co-V-2 infection at the beginning and throughout the study. We made corrections and added information about vaccination in the text.

Why group II have a vitamin D supplementation of 2000IU/day? If the authors aim to explore benefit of Vitamin D, why not compare to a group that with no vitamin D supplementation? Current setting is more like a comparison between high and low vitamin D supplementation.

Answer: Our center was involved in the treatment of COVID-19 patients only for three months, and hence the health care workers had direct contact with such patients only for this duration of time.  We do not have a control group since our Ethics Committee did not approve the control group among health care workers in contact with patients who had confirmed SARS-CoV-2 infection, those being as a high-risk of SARS-Co-V-2 infection group. Thus, group II received cholecalciferol at the common daily dose used in clinical practice – 2,000 IU/day.

  1. The Serum 25(OH)D level of these two groups seems mostly differ in the first two weeks (Figure 2).  Is there a significant difference of infection in this two-weeks period? 

Answer: In Group I there was no infected subjects and in Group II only one subject had positive PCR test in the two-week period from the start of treatment. There is no significant difference between groups in this two-week period. However, we did not check SARS-CoV-2 Ab in this period of the study.

Round 2

Reviewer 2 Report

The authors have addressed all my questions. I look forward to the publish of this study.